# Effects of a Theory-Based, Multicomponent eHealth Intervention for Obesity Prevention in Young Children from Low-Income Families: A Pilot Randomized Controlled Study

**DOI:** 10.3390/nu15102296

**Published:** 2023-05-13

**Authors:** Hyunjung Lee, Wilna Oldewage-Theron, John A. Dawson

**Affiliations:** 1Department of Nutrition, Texas A&M University, College Station, TX 77843, USA; 2Department of Nutritional Sciences, Texas Tech University, Lubbock, TX 79409, USA; wilna.oldewage@ttu.edu (W.O.-T.); dawsonja@nmsu.edu (J.A.D.); 3Department of Economics, Applied Statistics, and International Business, New Mexico State University, Las Cruces, NM 88003, USA

**Keywords:** eHealth, early childhood, obesity, prevention, healthy eating, active lifestyle, sedentary behavior, screen time, parents, feeding practices

## Abstract

The purpose of this study was to evaluate a theory-based, multicomponent eHealth intervention aimed at improving child health behaviors and parental psychosocial attributes and feeding practices. A pilot randomized controlled trial was conducted among 73 parents with children (1–3 years). Intervention group participants (IG, *n* = 37) received theory-based educational videos, cooking tutorials, and text messages with key information for a total of 8 weeks. Control group participants (CG, *n* = 36) received a booklet about general nutrition recommendations for children. A parent-administered questionnaire was used for data collection at baseline and post-intervention. Linear models were performed using R version 4.1.1. for data analysis. Children in the IG significantly increased their daily intake of fruit (ΔΔ = 0.89 servings, *p* = 0.00057) and vegetables (ΔΔ = 0.60 servings, *p* = 0.0037) and decreased use of screen time (ΔΔ = −33.87 min, *p* = 0.026), compared to the CG. Parents in the IG improved significantly more than the CG in self-efficacy (*p* = 0.0068) and comprehensive feeding practices (*p* = 0.0069). There were no significant differences between the study groups for changes in child outcomes, such as physical activity and sedentary behaviors, and parental nutrition knowledge and attitudes.

## 1. Introduction

In the United States (US), one in every five children (2–19 years) are obese [1]. In addition to the two-fold increase in childhood obesity over the past 20 years, obesity occurs disproportionately in children from less privileged socioeconomic backgrounds [2]. A meta-analysis of a nationally representative sample of children showed that low-income households are more prone to be overweight and obese than their high-income counterparts [2]. Households headed by parents with less than a high school degree have a considerably greater prevalence of childhood obesity than those with parental attainment of a college degree or higher [3]. Children of a single head-of-household are also more likely to be overweight than their peers of a dual head-of-household [4].

According to the Ecological Systems Theory, obesity risk factors for children interplay between three determinants: child characteristics, familial characteristics, and societal characteristics [5,6]. Beyond genetic predispositions, behavioral patterns, such as dietary choices, exercise, and sedentary behavior, can place a child at risk for obesity. The development of behavioral patterns in children is shaped by parenting styles and familial characteristics, such as parental nutrition knowledge, feeding practices, and family activity patterns. Finally, the development of childhood obesity is influenced by social contexts at large, such as the community, demographic, and wider environmental factors. For this study, parents were viewed as key mediators of change to reduce behavioral risk factors and promote healthy dietary and physical activity patterns in young children. 

Traditionally, face-to-face nutrition interventions have shown positive impacts on improving nutritional behavior changes in children [7]. One of the major strengths of these traditional interventions is the development of personal interactions between researchers and participants. However, there are major limitations in reaching individuals who have time and resource constraints, particularly low-income families who often work long hours and cannot afford transportation and gas money to access interventions that include regular face-to-face contact [8,9,10]. Furthermore, in a previous qualitative study by Zhen-Duan et al. [11], low-income parents reported that walking in their neighborhood and going outdoors is challenging because of safety concerns and extreme weather conditions. 

Web-based interventions are cost-effective and promising approaches for expanding an intervention’s reach to families with children. For example, Knowlden and Sharma [12] compared a website with healthy lifestyle information to a stand-alone intervention among parents with children aged 4 to 6 years. Results of this study showed an overall increase of 1.84 cups of vegetables and fruits among children in the intervention group, which was maintained at a one-year follow-up [12]. In addition, mHealth intervention using smartphones among parents of children aged 3 to 8 years old resulted in increased child intake of vegetables and fruits, as reflected in elevated skin carotenoid levels [13]. Yet, only a few studies have concentrated on the effectiveness of web-based interventions supporting parents of children from low-income backgrounds. Most online nutrition interventions also target children aged 3 years or older and do not incorporate parental feeding practices. However, research has shown that dietary patterns established as early as 1 year track into mid-childhood [14]. In addition, parental feeding practices can affect dietary patterns and risk of obesity in children. For example, coercive and highly protective feeding practices, such as pressure to eat, emotional regulation, and restriction of food practices, are associated with high consumption of snack foods and high body mass index (BMI) z scores in children [15,16]. 

This study evaluated the effectiveness of a theory-based, multicomponent eHealth intervention among low-income families of young children (1–3 years). The primary study objectives were to increase daily vegetable and fruit intake and physical activity (i.e., active playtime) and reduce sedentary behavior and screen time among children. Secondarily, this study aimed to improve nutrition knowledge, self-efficacy in feeding, attitudes towards healthy eating and physical activity, and feeding practices among parents. The promotion of responsive feeding practices is integrated within the eHealth intervention to support parents in achieving healthy feeding goals.

## 2. Materials and Methods

### 2.1. Research Design

A pilot randomized control and intervention study was conducted in the South Plains of Texas between October 2021 and February 2022. The study was registered at Clinicaltrials.gov (Identifier: NCT05085041, accessed on 7 May 2023). The intervention was an 8-week eHealth program, combined with a provision of fresh fruit and vegetables biweekly, to encourage parents to improve their children’s eating and physical activity habits.

### 2.2. Participants

Parents with children aged between 1 and 3 years were recruited at 4 childcare centers in the South Plains of Texas. To reach the intended audience, evidence-based three-step recruitment strategies were used: (1) promoting the study before formal requests to participate, (2) disseminating informational packages with consent forms, and (3) reminding parents about the study and consent forms [17]. Throughout the recruitment steps, digital tools, such as newsletters and text messages, were used to expand the reach among working parents and families who registered for home-based services. Parents who were main food preparers in the home, physically and mentally capable of communicating, competent in reading and writing in English, and willing to attend the online intervention were eligible for the study. 

A total of 73 eligible participants were enrolled in the study (Figure 1). Prior to the study, all participants were informed about the study’s purpose, procedures, risks, and benefits. After acquiring informed consent forms, participants were randomly assigned to the intervention (*n* = 37) or control group (*n* = 36) via block randomization. The statistical software R version 4.1.1. was used to generate randomization codes of a block size of 4, stratifying by location of study recruitment. The intervention group completed the 8-week eHealth program, including educational videos, cooking tutorials, and reminder text messages. Participants in the control group received a booklet about general nutrition recommendations for children after completing baseline measurements. All participants received USD 30 gift cards upon the completion of post-intervention measurements. The study was given ethical approval by Texas Tech University’s Institutional Review Board (IRB2021-505). 

### 2.3. Intervention Methods

The researcher developed the Website™ (Google LLC, Mountain View, CA, USA) to facilitate easy access to intervention content among participants. The intervention group received weekly educational videos through the developed website and weekly reminder text messages with key information for a total of 8 weeks. The educational videos included pertinent visuals and subtitles to improve video quality, engagement, and participants’ understanding. The researcher instructed participants to respond to reflective questions that were sporadically inserted into the videos in order to track intervention adherence. 

Social Cognitive Theory (SCT) is the commonly used theory in behavioral change interventions viewing parents as key change agents to improve child obesogenic behavioral factors [7,18,19]. In this study, the development of the intervention curriculum and activities was informed by SCT to build parental knowledge, self-efficacy, and skills related to a healthy lifestyle (Table 1). For example, parents were instructed to set parental feeding goals using Specific, Measurable, Achievable, Realistic, and Time-framed (SMART) principles at the beginning of the intervention. These goals were intended to enhance self-efficacy, autonomy, and health behavior changes in parents [20,21]. The researcher also encouraged parents to involve family members in the SMART goal-setting process. This may have allowed opportunities for other family members to understand the benefits of healthy eating and increased physical exercise. Participants reflected on their SMART goals once every two weeks as an opportunity to recognize small accomplishments on the way to larger behavior change and to receive personalized feedback from the researcher. In addition, intervention group parents received online cooking tutorials to create budget-friendly and healthy meals and snacks, using fruit and vegetables, for their children. Food ingredients and utensils (My ChildPlate) were provided to the intervention group as encouraging reinforcements to complete cooking tutorial activities. 

### 2.4. Survey Items

A parent-administered questionnaire was used to collect household sociodemographic characteristics, such as child age, child ethnicity/race, relationship with the child, parental educational attainment and marital status, and yearly household income. Child and parental anthropometric data were collected in kg and cm using a portable digital Omron scale and stadiometer height equipment [22]. The researcher followed the Centers for Disease Control and Prevention (CDC)’s guidelines to measure anthropometry. For example, toddlers who were unable to stand on the scale by themselves were encouraged to be weighed with the help of a parent [23]. In this case, the researcher required the parent and child to be weighed simultaneously, then the weight of the parent was subtracted from the total weight. The researcher asked the parent and child to remove their shoes, bulky clothing, and any hair accessories for accurate height and weight measurements [24]. 

Anthropometric data were then converted into BMI-for-age (for toddlers) and BMI (for parents) to screen overweight, obese, and underweight. For children aged between 1 and 2 years, the CDC recommends using the WHO growth charts [25]. Weights and heights of children of this age range were converted to BMI-for-age z scores using WHO Anthro version 3.2.2. For children aged between 2 and 3 years, BMI-for-age z scores were calculated using the CDC’s BMI calculator [26]. Children with BMI-for-age z scores <−2.0 were considered as underweight, >2.0 as overweight, and >3.0 as obese [27]. Lastly, parental BMI was calculated as “weight in kg divided by height in meters squared” [28]. Parents with BMI between 18.5 and 24.9 were considered a healthy weight, 25 and 29.9 as overweight, and BMI equal to or greater than 30 as obese [28].

#### 2.4.1. Primary Evaluation Items

Parents used mobile devices to capture digital food photos of their 24-h child’s meals, snacks, and beverages at baseline and post-intervention. Child fruit (excluding fruit juices) and vegetable intakes (excluding French fry products) were recorded in daily servings based on food photos. The researcher provided a face-to-face demonstration of how to record food photos and estimate serving sizes offered to and consumed by children. Parents sent food photos in real-time and written descriptions, such as serving sizes, cooking methods, and food labels. 

The Early Years Physical Activity Questionnaire (EY-PAQ) was used to evaluate habitual moderate-to-vigorous physical activity, sedentary behavior, and screen time in children [28]. Parents were asked to report the frequency and duration in which their child was engaged in different moderate-to-vigorous physical activities (e.g., playing in the park, playground, and indoor play facilities), sedentary behavior (e.g., coloring/drawing, reading/being read to, and traveling in a car or pushchair), and screen time (e.g., watching television or videos during meals or not during meals and playing a non-active computer/tablet/cellphone game) at baseline and post-intervention.

#### 2.4.2. Secondary Evaluation Items

Parental psychosocial attributes of nutrition knowledge, attitudes, and self-efficacy were measured at baseline and post-intervention. A multiple-choice nutrition knowledge questionnaire was developed for use in this study with a possible score range from 0 to 20. Question items asked parents to choose the correct answer to questions about healthy eating and physical activity principles for young children (1–3 years), consistent with the 2020–2025 dietary guidelines by the US Department of Agriculture [29] and international physical activity guidelines by US Department of Health and Human Services [30]. Parental attitudes towards healthy eating and physical activity were assessed using a 13-item questionnaire with responses on a 5-point Likert scale [31]. Responses of “strongly disagree” were coded as 0 and “strongly agree” as 4. A total score of parental attitudes ranged from 0 to 52, with higher scores indicating positive attitudes toward children’s healthy eating and physical activity. The self-efficacy questionnaire of 8 question items was used to gauge how effective parents felt they were at feeding their children and interacting with them during meals [32]. Responses of “not confident at all” were coded as 0 and “very confident” as 4, with the possible total score range from 0 to 32. 

A comprehensive range of parental feeding practices was assessed at baseline and post-intervention using the Comprehensive Feeding Practices Questionnaire [33]. On a 5-point Likert scale (“never, rarely, sometimes, mostly, always”), parents expressed their level of agreement with the frequency of utilizing responsive and non-responsive parental feeding practices. Responsive feeding practices included encouragement (4 question items), healthy environment (4 question items), involvement of the child in meal planning and preparation (3 question items), modeling (4 question items), and monitoring of children’s food consumption (4 question items). For the responsive feeding practices question items, responses of “never” were coded as 0 and “always” as 4. Non-responsive feeding practices included child control (5 question items), using food for emotion regulation (3 question items), food as a reward (3 question items), pressure to eat (4 question items), restriction for health (4 question items), and restriction for weight (8 question items. Responses for the non-responsive feeding practices were reverse coded (“always” as 0 and “never” as 4). The total sum score for comprehensive parental feeding practices ranged from 0 to 184, with a higher score reflecting overall healthy parental feeding practices. 

### 2.5. Statistical Analysis

Descriptive statistics were used to describe the background characteristics of participants. Baseline values for primary and secondary outcomes were compared to check group equivalences between the intervention and control groups, using the Mann–Whitney U test and Fisher’s exact test. Linear models were performed to assess changes in primary and secondary study outcomes within and between the study groups, from baseline to post-intervention. Variables that were significantly different between the study groups at baseline and the location of recruitment sites were considered potential covariates in initial models but dropped if they were nonsignificant. Baseline outcome values were also adjusted to account for potential floor or ceiling effects. The software R version 4.1.1 was used for data analysis. 

## 3. Results

### 3.1. Background Characteristics

A total of 73 parents with children were included in this study. The majority (56.2%) of children were female, with an average age of 26.51 ± 8.48 months. Twenty-seven (36.9%) children were biracial, followed by Hispanic (23.3%), Non-Hispanic White (21.9%), Black (16.4%), and Native Hawaiian and other Pacific Islanders (1.4%). About one-third of children were overweight (21.9%) or obese (13.7%). The average BMI-for-age z scores of children were 1.03 ± 1.88 and 0.71 ± 1.41 in the intervention and control groups, respectively. There were no significant differences in child demographic and anthropometric characteristics between the study groups (Table 2). 

Most parents were mothers (84.9%), and only 20.5% obtained a bachelor’s degree or higher. Almost half of the parents were never married (49.3%). The average yearly household income was USD 26,436.97 ± 17,524.52 among the participating families. The majority of households had very low (56.1%) or low (13.7%) household income levels. The average parental BMI was 31.71 ± 8.34, with most participants being obese (46.6%) or overweight (34.2%). Parental BMI was significantly higher in the control group. No significant differences were found in other parental and household characteristics between the study groups at baseline (Table 2). A comparison of background characteristics with dropouts is included in Appendix A.

### 3.2. Results of Child Eating and Physical Activity Behaviors

The mean change of daily fruit intake (servings/day) within the intervention group significantly increased from 0.57 to 1.47 (Δ = 0.91, *SD*_Δ_ = 0.62; *t* (30.31) = 8.89, *p* = 0.0000000006). There was no change in the mean fruit intake among children in the control group (from 0.47 to 0.46, Δ = −0.016, *SD*_Δ_ = 0.88; *t* (21.54) = −0.11, *p* = 0.92). Linear models showed that children in the intervention group had a significant increase in fruit intake at post-intervention, compared to the control group (ΔΔ = 0.89, *SD*_ΔΔ_ = 1.93; *t* (24.69) = 3.96, *p* = 0.00057). While control group children had no change in daily vegetable intake (from 0.44 to 0.27, Δ = −0.17, *SD*_Δ_ = 0.74; *t* (22.58) = −1.40, *p* = 0.17), the intervention group showed a small but significant increase from 0.45 to 0.98 (Δ = 0.54, *SD*_Δ_ = 0.69; *t* (29.39) = 4.74, *p* = 0.00005). This increase in vegetable intake among the intervention group was significant compared to the control group (ΔΔ = 0.60, *SD*_ΔΔ_ = 1.64; *t* (36.13) = 3.10, *p* = 0.0037).

Children in the control group had no changes in daily minutes of moderate-to-vigorous physical activity (from 61.23 to 55.58, Δ = −5.65, *SD*_Δ_ = 39.29; *t* (23.88) = −0.86, *p* = 0.4). The intervention group significantly increased moderate-to-vigorous physical activity at post-intervention (from 59.79 to 69.07, Δ = 9.28, *SD*_Δ_ = 27.15; *t* (27.23) = 2.08, *p* = 0.047). However, this increase did not remain significant as compared to the control group (ΔΔ = 10.13, *SD*_ΔΔ_ = 81.45; *t* (22.52) = 1.06, *p* = 0.29). Daily minutes of sedentary behavior significantly decreased among children in the intervention group (from 238.33 to 194.96, Δ = −43.38, *SD*_Δ_ = 61.79; *t* (30.34) = −4.27, *p* = 0.00018), whereas the control group had no changes in sedentary behavior (from 208.38 to 238.15, Δ = 29.78, *SD*_Δ_ = 97.59; *t* (21.66) = 1.83, *p* = 0.08). Results of linear models showed no significant differences between the groups in sedentary behavior at post-intervention (ΔΔ = −44.89, *SD*_ΔΔ_ = 191.86; *t* (32.36) = −2.00, *p* = 0.054). Intervention group children showed a significant reduction in screen time (minutes/day) from 73.25 to 45.78 (Δ = −27.47, *SD*_Δ_ = 30.87; *t* (30.15) = −5.41, *p* = 0.000007). The control group had no change in screen time (from 68.31 to 89.45, Δ = 21.14, *SD*_Δ_ = 49.12; *t* (23.38) = 2.58, *p* = 0.017). The mean change in daily screen time among children in the intervention group was significant compared to the control group (ΔΔ = −33.87, *SD*_ΔΔ_ = 121.67; *t* (23.05) = −2.37, *p* = 0.026). Table 3 summarizes the results of the pre-post changes in child outcomes. Baseline value comparisons of child eating and physical activity behaviors with dropouts are shown in Appendix A. 

### 3.3. Results of Parental Psychosocial Attributes and Feeding Practices

Parental nutrition knowledge did not show significant changes within the intervention (from 10.89 to 11.49, Δ = 0.60, *SD*_Δ_ = 2.33; *t* (30.43) = 1.57, *p* = 0.13) and control groups (from 11.11 to 10.34, Δ = −0.77, *SD*_Δ_ = 2.62; *t* (22.13) = −1.76, *p* = 0.093). There were no differences in change scores of parental nutrition knowledge between the study groups at post-intervention (ΔΔ = 0.92, *SD*_ΔΔ_ = 6.37; *t* (29.99) = 1.23, *p* = 0.23) (Table 4). Baseline value comparisons of parental psychosocial attributes with dropouts are shown in Appendix A.

Intervention group parents significantly improved their scores of attitudes toward children’s healthy eating and physical activity (from 33.51 to 36.20, Δ = 2.68, *SD*_Δ_ = 5.94; *t* (29.20) = 2.74, *p* = 0.01). The control group had no changes in their scores of attitudes (from 34.08 to 33.18, Δ = −0.90, *SD*_Δ_ = 6.63; *t* (26.38) = −0.82, *p* = 0.42). There were no significant differences between the study groups in change scores of parental attitudes at post-intervention (ΔΔ = 2.68, *SD*_ΔΔ_ = 16.14; *t* (32.78) = 1.42, *p* = 0.17). 

The intervention group had a significant increase in self-efficacy levels post-intervention (from 22.78 to 27.35, Δ = 4.57, *SD*_Δ_ = 4.66; *t* (31.04) = 5.96, *p* = 0.000001). The control group had no changes in self-efficacy (from 25.25 to 24.19, Δ = −1.06, *SD*_Δ_ = 5.08; *t* (21.83) = −1.25, *p* = 0.22). The increased change score of self-efficacy level among the intervention group was significant compared to the control group (ΔΔ = 4.42, *SD*_ΔΔ_ = 12.94; *t* (28.57) = 2.92, *p* = 0.0068). 

The mean score of comprehensive feeding practices showed a significant increase in the intervention group (from 123.19 to 132.09, Δ = 8.90, *SD*_Δ_ = 20.23; *t* (30.30) = 2.68, *p* = 0.012). For example, intervention group parents reported improved changes in their levels of agreement with encouragement (from 13.24 to 15.20, Δ = 1.95, *SD*_Δ_ = 2.56; *t* (31.24) = 4.63, *p* = 0.00006), involvement (from 6.27 to 9.35, Δ = 3.08, *SD*_Δ_ = 3.61; *t* (27.20) = 5.19, *p* = 0.00002), modeling (from 11.89 to 13.76, Δ = 1.87, *SD*_Δ_ = 2.39; *t* (29.37) = 4.75, *p* = 0.00005), emotion regulation (from 8.00 to 9.17, Δ = 1.17, *SD*_Δ_ = 2.64; *t* (30.43) = 2.69, *p* = 0.012), food as a reward (from 7.32 to 9.29, Δ = 1.96, *SD*_Δ_ = 3.28; *t* (23.42) = 3.65, *p* = 0.0013), pressure to eat (from 7.19 to 9.09, Δ = 1.90, *SD*_Δ_ = 3.36; *t* (31.99) = 3.45, *p* = 0.0016), and restriction for weight (from 23.78 to 26.73, Δ = 2.95, *SD*_Δ_ = 4.61; *t* (29.61) = 3.89, *p* = 0.00053) (Table 5). The control group showed a significant decrease in the mean score of comprehensive feeding practices (from 132.19 to 120.70, Δ = −11.50, *SD*_Δ_ = 15.45; *t* (21.87) = −4.46, *p* = 0.002). There was a significant difference in the mean score of comprehensive feeding practices between the study groups at post-intervention (ΔΔ = 13.69, *SD*_ΔΔ_ = 40.89; *t* (37.46) = 2.85, *p* = 0.0069). Correlational results between the eHealth intervention attainment among parents and changes in child health behaviors are shown in Appendix A. 

## 4. Discussion

This study addressed the effectiveness of a theory-based, multicomponent eHealth intervention on fruit and vegetable intake, moderate-to-vigorous physical activity, sedentary behavior, and screen time among young children (1–3 years). Secondarily, this study investigated if there were any improvements in nutrition knowledge, attitudes, self-efficacy, and comprehensive feeding practices among their parents. 

Results of this study showed that intervention group children had significant improvements in fruit and vegetable intake by 0.91 and 0.54 servings/day, respectively, over the course of two months. Such increases in child fruit and vegetable intake in the intervention group remained significant when compared to the control group. Similar to the present study, Hunsaker and colleagues conducted a randomized controlled trial among 65 parents with preschoolers and kindergartners in Utah [34]. Parents in the intervention group from the Utah study received a parent health report via email about fruit and vegetable recommendations for their child [34]. Children in the intervention group significantly increased their vegetable intake by 0.65 servings/day, which is comparable to our findings [34]. Although it targeted older children aged three to eight years old, Bakırcı-Taylor et al. [13] examined the feasibility of the mobile Jump2Health website, where parents received information about healthy eating and physical activity for their children, via a Facebook page for 10 weeks. Results of this study showed significant improvements in skin carotenoid levels among children and parents in the intervention group than the control group. 

In this study, intervention group children increased moderate-to-vigorous physical activity. However, this increase was not significant compared to the control group. Similar results were found in another study among preschoolers whose parents received a 10-week lifestyle intervention via Facebook, as well as a randomized controlled trial for early childhood health promotion among parents with preschoolers [35,36]. A more recent randomized controlled trial study by Ha et al. [37] examined the effectiveness of a family-based intervention called Active 1 + FUN. Similar to our findings, immediately post-intervention, they did not find significant differences in daily minutes of moderate-to-vigorous physical activity in children between intervention and control groups [37]. It may be possible that there was a small change because of already adequate levels of moderate-to-vigorous physical activity reported at baseline among participating children in the present study. In addition, our study focused on intervention strategies to help parents understand the health benefits of MVPA for young children and apply affordable at-home physical exercise options. Promoting different types of MVPA, instead of the intensity of such activities, may contribute to nonsignificant results. In contrast to our nonsignificant results, a previous randomized controlled trial among preschools, which focused on increasing the intensity of outdoor play in school settings, found significant increases in MVPA of 7.3 min/h in the intervention group [38]. 

Significant intervention effects on child sedentary behavior and screen time were observed in this study. Consistent with our findings, a systematic review of randomized controlled trials found a significant intervention effect in reducing sedentary time by an average of 18.91 min/day and screen time by 17 min/day among children aged 2 to 5 years [39]. Early childhood may be an opportune time to intervene in changing dietary intake, sedentary time, and screen time. However, there are few interventions designed to reduce sedentary behavior and screen time in children under three and outside the preschool setting, suggesting that further research is needed [40]. 

Parents in this study started with relatively high knowledge and attitude scores. For example, at baseline, the intervention and control group answered correctly to an average of 57.9% and 57.3% for knowledge question items and 65.4% and 63.5% for attitude question items. Higher baseline scores generally leave less room for improvement, which may contribute to an insignificant magnitude of change. In contrast to knowledge and attitudes, parents showed significant improvements in self-efficacy around feeding. Consistent with our findings, another study examining the effect of an online video intervention among parents with children aged 6 to 12 years found a significant effect on parental self-efficacy post-intervention [41]. Contrasting results were found in a randomized controlled trial study among parents with kindergarteners, where it found nonsignificant intervention effects on parental self-efficacy, possibly due to not having active but only minimal and passive parental involvement [42]. 

This study observed significant improvements in comprehensive parental feeding practices. For example, intervention group parents showed significant changes in their levels of agreement with utilizing less non-responsive feeding practices, such as emotion regulation, food as a reward, pressure to eat, and restriction for weight post-intervention. Such coercive, protective feeding practices are found to have unintentional outcomes in children. A systematic review found a significant association between controlling/restrictive parental feeding practices with high BMI-for-age z scores and, thus, obesity risk in young children [15]. Similarly, increased consumption of processed snack foods in young children is linked to emotional regulation feeding, which is when parents give food to their children in response to their child’s negative feelings (e.g., boredom and distress) [16]. Hammersley et al. [43] found significant changes in parental use of pressure to eat as a result of an eHealth program among parents with preschoolers in Australia. A standard face-to-face intervention, using a weekly newsletter on child feeding situations, also led to significant changes in parental use of pressure to eat [44]. Similarly, a home-based obesity prevention study found that parents were engaged in less restrictive feeding practices post-intervention [45]. However, most interventional studies measured only small subsets of parental feeding practices. Given the complex relationships between parental feeding practices and child obesity risks, the more frequent inclusion of a broad range of feeding practices should be considered when designing and evaluating childhood obesity prevention programs.

Interventions using digital methods have several notable strengths. Our intervention website was a user-friendly stand-alone product that has the potential for widespread application and sizable reach. Additionally, previous digital behavioral change interventions focused solely on parental psychosocial attributes regarding child obesogenic health behaviors. A unique aspect of this study is its holistic approach to enhancing parenting skills through awareness of responsive feeding practices, food preparation skills through cooking tutorials, and parental intentions through improved parental psychosocial determinants. This study facilitated health behavior changes in young children at home. Relatively few studies were undertaken among 1–3 year-old children and outside the preschool setting. Child screen time behavior also has not been considered in many early interventions. However, this study is not without some limitations. The researcher did not have follow-up measurements; thus, the maintenance of behavior changes in participants over time cannot be explained. Parent-administered proxy questionnaires were used to assess study outcomes, and therefore, there may be reporting bias among participants. The lack of standardized methods for eating and behavioral patterns remains a major limitation and challenge. This study did not track BMI-for-age for children or BMI for parents, so the impact of the intervention on actual weight status cannot be explained.

## 5. Conclusions

Results of this study show that empowering families through a theory-based, multicomponent eHealth intervention program is a promising approach for improving nutrition- and health-related behavioral changes in young children. Our study sample was mostly from underprivileged socioeconomic backgrounds who are susceptible to low food security and future obesity. eHealth interventions may be an effective tool to help reduce childhood obesity disparities by developing healthy dietary and activity patterns in children from low-income backgrounds. Further investigation with a follow-up assessment is needed to demonstrate the effectiveness of web-based programs on the maintenance of behavioral changes.

## Figures and Tables

**Figure 1 nutrients-15-02296-f001:**
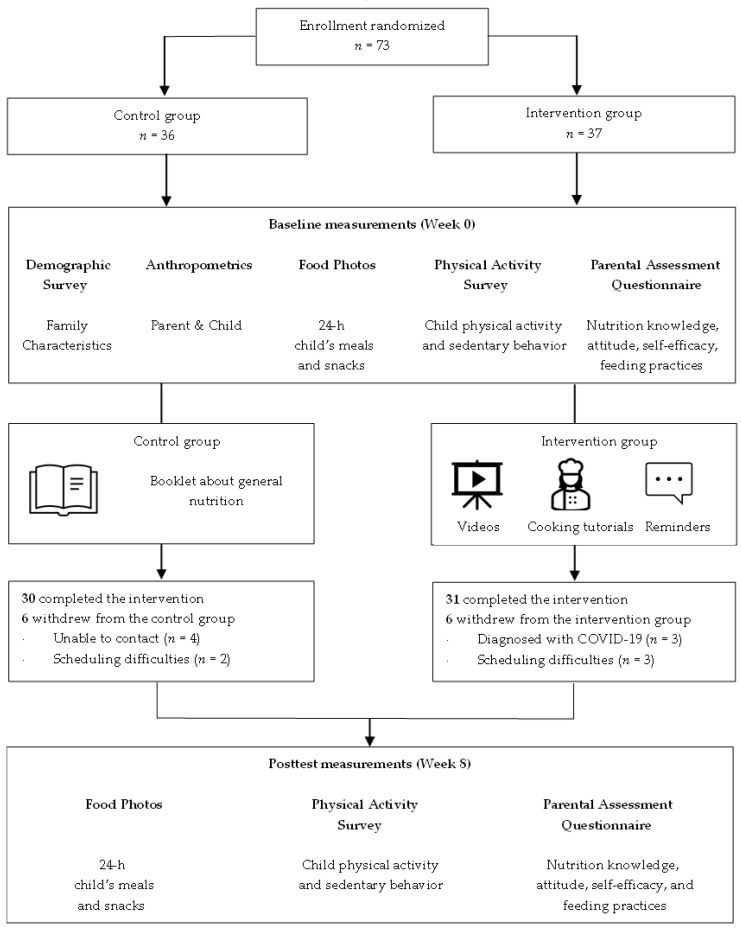
Participant flow diagram.

**Table 1 nutrients-15-02296-t001:** eHealth intervention curriculum and activities.

Session	Topic	Activity
1	What Does Healthy Eating Look Like for Children	
	Introduction; food groups using MyPlate; trends and consequences of childhood obesity; portion sizes	Setting SMART goal
2	Why Children Should Eat More Fruit	
	Recommended fruit intake; health benefits of eating various fruit; tips on introducing new fruits to toddlers	Making healthy snacks using fruits
3	Listening to Your Child’ Needs	
	Importance of using responsive feeding practices; impact of growth and psychosocial development on child appetite; neophobia or rejection of foods	Preparing child meals using ‘My ChildPlate’
4	How to Practice Responsive Feeding	
	Emotional, pressure, and restriction feeding practices; negative consequences of such practices in children; tips on shifting non-responsive to responsive feeding practices	Journaling on current feeding practices
5	Vegetable Goals for Young Children	
	Recommended vegetable intake; health benefits of eating adequate and various vegetables; tips on increasing vegetable intake at home; importance of parent modeling in eating vegetables	Making veggie stuffed pepper cups
6	Active and Healthy Child	
	Physical activity and screen viewing guidelines for toddlers; examples and strategies for increasing physical activity at home without equipment	Identifying negative consequences of child screen time
7	Eating on A Budget	
	Tips to shop on a budget for fruit and vegetables (both perishable and non-perishable items); importance of meal planning; utilizing leftovers	Making healthy tortilla pizza
8	Informed Shopper	
	Elements on food labels; health claims; serving size; what and how to read food labels	Reading food labels

**Table 2 nutrients-15-02296-t002:** Baseline characteristics of participants (*n* = 73).

Characteristic	Full Sample	Intervention Group	Control Group	*p*-Value
Age (months)	26.52 ± 8.48	25.54 ± 8.67	27.53 ± 8.28	0.26
Gender				0.58
Female	41 (56.2%)	19 (51.4%)	13 (36.1%)	
Male	32 (43.8%)	18 (48.7%)	23 (63.9%)	
Race				0.48
Biracial	27 (36.9%)	14 (37.8%)	13 (36.1%)	
Hispanic	17 (23.3%)	10 (27.0%)	6 (16.7%)	
Non-Hispanic White	16 (21.9%)	6 (16.2%)	11 (30.6%)	
Black	12 (16.4%)	7 (18.9%)	5 (13.9%)	
Native Hawaiian and other Pacific Islander	1 (1.4%)	0	1 (2.8%)	
BMI-for-age z score	0.87 ± 1.65	1.03 ± 1.88	0.71 ± 1.41	0.504
Weight status				0.50
Healthy weight	41 (56.2%)	24 (64.9%)	17 (47.2%)	
Obesity	16 (21.9%)	7 (18.9%)	9 (25.0%)	
Overweight	10 (13.7%)	4 (10.8%)	6 (16.7%)	
Underweight	6 (8.2%)	2 (5.4%)	4 (11.1%)	
Relationship with child				0.83
Mother	62 (84.9%)	32 (86.5%)	30 (83.3%)	
Father	6 (8.2%)	2 (5.4%)	4 (11.1%)	
Foster mother	4 (5.5%)	2 (5.4%)	2 (5.6%)	
Grandmother	1 (1.4%)	1 (2.7%)	0	
Parental educational attainment				0.21
High school	22 (30.1%)	14 (37.8%)	8 (2.2%)	
One year of college	13 (17.8%)	6 (16.2%)	7 (19.4%)	
Bachelor’s degree or equivalent	10 (13.7%)	6 (16.2%)	3 (8.3%)	
Two years of college	9 (12.3%)	5 (13.5%)	4 (11.1%)	
Some high school	9 (12.3%)	4 (10.8%)	6 (16.7%)	
Three years of college	5 (6.8%)	2 (5.4%)	3 (8.3%)	
Master’s degree	5 (6.8%)	0	5 (13.9%)	
Parental marital status				0.47
Never married	36 (49.3%)	19 (51.4%)	17 (47.2%)	
Married	25 (34.2%)	14 (37.8%)	11 (30.6%)	
Divorced	7 (9.6%)	2 (5.4%)	5 (13.9%)	
Separated	3 (4.1%)	2 (5.4%)	1 (2.8%)	
Engaged	2 (2.7%)	0	2 (5.6%)	
Yearly income ($)	26,436.97 ± 17,524.52	26,113.43 ± 17,536.31	26,808.44 ± 17,837.48	0.94
Household income level				0.91
Very low income	41 (56.1%)	21 (56.8%)	20 (5.6%)	
Low income	10 (13.7%)	6 (16.2%)	4 (11.1%)	
Non-low income	4 (5.5%)	1 (2.7%)	3 (8.3%)	
Not specify	18 (24.7%)	9 (24.3%)	9 (25.0%)	
Parental BMI	31.71 ± 8.34	29.95 ± 8.35	33.43 ± 8.07	0.03
Parental weight status				0.20
Obesity	34 (46.6%)	21 (56.8%)	13 (36.1%)	
Overweight	25 (34.2%)	11 (29.7%)	14 (38.9%)	
Healthy weight	14 (19.2%)	5 (13.5%)	9 (25.0%)	

**Table 3 nutrients-15-02296-t003:** Pre-post changes in child health behaviors within and between study groups (*n* = 73).

Variable	Control Group (*n* = 36)					Intervention Group (*n* = 37)					Between Groups		
	Baseline M [SD]	Posttest M [SD]	Change Score M [SD]	t [df]	*p*-Value	BaselineM [SD]	Posttest M [SD]	Change Score M [SD]	t [df]	*p*-Value	Change Score M [SD]	t [df]	*p*-Value
Fruit intake (servings)	0.47 [0.62]	0.46 [0.75]	−0.016 [0.88]	−0.11 [21.54]	0.92	0.57 [0.48]	1.47 [0.54]	0.91 [0.62]	8.89 [30.31]	6 × 10^−10^	0.89 [1.93]	3.96 [24.69]	0.00057
Vegetable intake (servings)	0.44 [0.47]	0.27 [0.66]	−0.17 [0.74]	−1.40 [22.58]	0.17	0.45 [0.52]	0.98 [0.71]	0.54 [0.69]	4.74 [29.39]	5 × 10^−5^	0.60 [1.64]	3.10 [36.13]	0.0037
MVPA (minutes)	61.23 [38.13]	55.58 [39.00]	−5.65 [39.29]	−0.86 [23.88]	0.4	59.79 [25.22]	69.07 [29.01]	9.28 [27.15]	2.08 [27.23]	0.047	10.13 [81.45]	1.06 [22.52]	0.29
Sedentary behavior (minutes)	208.38 [72.56]	238.15 [93.21]	29.78 [97.59]	1.83 [21.66]	0.08	238.33 [86.98]	194.96 [74.92]	−43.38 [61.79]	−4.27 [30.34]	0.00018	−44.89 [191.86]	−2.00 [32.26]	0.054
Screen time (minutes)	68.31 [48.58]	89.45 [63.98]	21.14 [49.12]	2.58 [23.38]	0.017	73.25 [34.74]	45.78 [33.34]	−27.47 [30.87]	−5.41 [30.15]	7 × 10^−6^	−33.87 [121.67]	−2.37 [23.05]	0.026

Note. MVPA: Moderate-to-Vigorous Physical Activity.

**Table 4 nutrients-15-02296-t004:** Pre-post changes in parental psychosocial attributes and comprehensive feeding practices within and between study groups (*n* = 73).

Variable	Control Group (*n* = 36)					Intervention Group (*n* = 37)					Between Groups		
	Baseline M [SD]	Posttest M [SD]	Change Score M [SD]	t [df]	*p*-Value	Baseline M [SD]	Posttest M [SD]	Change Score M [SD]	t [df]	*p*-Value	Change Score M [SD]	t [df]	*p*-Value
Nutrition knowledge	11.11 [1.74]	10.34 [2.58]	−0.77 [2.62]	−1.76 [22.13]	0.093	10.89 [1.71]	11.49 [2.50]	0.60 [2.33]	1.57 [30.43]	0.13	0.92 [6.37]	1.23 [29.99]	0.23
Attitude	34.08 [6.83]	33.18 [8.24]	−0.90 [6.63]	−0.82 [26.38]	0.42	33.51 [5.65]	36.20 [7.31]	2.68 [5.94]	2.74 [29.20]	0.01	2.68 [16.14]	1.42 [32.78]	0.17
Self-efficacy	25.25 [5.16]	24.19 [5.94]	−1.06 [5.08]	−1.25 [21.83]	0.22	22.78 [5.02]	27.35 [5.01]	4.57 [4.66]	5.96 [31.04]	1 × 10^−6^	4.42 [12.94]	2.92 [28.57]	0.0068
CFP	132.19 [15.71]	120.70 [17.99]	−11.50 [15.45]	−4.46 [21.87]	0.002	123.19 [21.39]	132.09 [18.07]	8.90 [20.23]	2.68 [30.30]	0.012	13.69 [40.89]	2.85 [37.46]	0.0069

Note. CFP: Comprehensive Feeding Practices.

**Table 5 nutrients-15-02296-t005:** Pre-post changes in subscales of comprehensive feeding practices (*n* = 73).

Variable	Control Group (*n* = 36)					Intervention Group (*n* = 37)					Between Groups		
	Baseline M [SD]	Posttest M [SD]	Change Score M [SD]	t [df]	*p*-Value	Baseline M [SD]	Posttest M [SD]	Change Score M [SD]	t [df]	*p*-Value	Change Score M [SD]	t [df]	*p*-Value
Responsive Feeding Practices													
Encouragement	13.86 [2.29]	13.79 [2.92]	−0.072 [2.24]	−0.19 [27.07]	0.84	13.24 [2.36]	15.20 [2.26]	1.95 [2.56]	4.63 [31.24]	6 × 10^−5^	0.98 [5.98]	1.39 [32.44]	0.17
Environment	10.50 [3.20]	10.65 [3.71]	0.15 [2.89]	0.30 [22.48]	0.76	10.43 [3.45]	11.53 [4.11]	1.10 [4.34]	1.54 [31.38]	0.13	1.18 [9.31]	1.08 [27.67]	0.29
Involvement	7.69 [2.95]	7.13 [3.59]	−0.56 [2.71]	−1.25 [26.70]	0.22	6.27 [3.20]	9.35 [3.45]	3.08 [3.61]	5.19 [27.20]	2 × 10^−5^	2.16 [10.51]	1.75 [19.04]	0.096
Modeling	12.92 [4.03]	12.44 [5.20]	−0.48 [3.31]	−0.86 [17.26]	0.41	11.89 [3.44]	13.76 [3.04]	1.87 [2.39]	4.75 [29.37]	5 × 10^−5^	1.61 [6.60]	2.09 [41.99]	0.043
Monitoring	11.08 [4.20]	12.61 [5.44]	1.53 [5.28]	1.74 [22.74]	0.096	11.24 [4.41]	12.10 [4.10]	0.85 [4.95]	1.05 [32.06]	0.30	−0.77 [11.04]	−0.60 [31.94]	0.56
Non-Responsive Feeding Practices													
Child control	11.39 [3.56]	10.11 [4.14]	−1.28 [3.76]	−2.04 [27.46]	0.051	10.05 [2.86]	10.60 [4.37]	0.54 [3.86]	0.85 [25.25]	0.4	1.20 [9.47]	1.08 [33.04]	0.29
Emotion regulation	8.31 [2.54]	8.80 [3.19]	0.49 [2.95]	1.00 [20.54]	0.33	8.00 [2.33]	9.17 [2.41]	1.17 [2.64]	2.69 [30.43]	0.012	0.84 [6.21]	1.16 [27.28]	0.26
Food as reward	8.14 [2.50]	7.12 [4.28]	−1.01 [4.22]	−1.44 [22.03]	0.16	7.32 [2.84]	9.29 [3.17]	1.96 [3.28]	3.65 [23.42]	0.0013	2.23 [10.25]	1.86 [24.05]	0.075
Pressure to eat	7.44 [3.74]	6.40 [4.60]	−1.04 [4.94]	−1.27 [24.54]	0.22	7.19 [3.82]	9.09 [3.41]	1.90 [3.36]	3.45 [31.99]	0.0016	2.51 [11.80]	1.82 [21.84]	0.083
Restriction for health	6.89 [3.98]	6.05 [4.87]	−0.84 [4.07]	−1.24 [21.78]	0.23	6.57 [2.52]	7.71 [4.24]	1.14 [3.53]	1.96 [27.85]	0.06	1.98 [9.44]	1.79 [31.17]	0.083
Restriction for weight	26.53 [4.08]	25.70 [5.72]	−0.82 [5.08]	−0.97 [20.20]	0.34	23.78 [4.81]	26.73 [6.55]	2.95 [4.61]	3.89 [29.61]	0.00053	2.16 [13.39]	1.38 [26.02]	0.18

## Data Availability

The data presented in this study are available on reasonable request from the corresponding author: grace.lee@ag.tamu.edu.

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
