# Peer review of "Effects of a Theory-Based, Multicomponent eHealth Intervention for Obesity Prevention in Young Children from Low-Income Families: A Pilot Randomized Controlled Study"

_nutrients, 2023, doi:10.3390/nu15102296_

Round 1
Reviewer 1 Report
Dear Editor,
After reading Research Article titled " Effects of a Theory-Based, Multicomponent eHealth Intervention for Obesity Prevention in Young Children from Low-Income Families: A Pilot Randomized Controlled Study", I recommend that it should be revised taking into account following requested and comments.
1. Authors claim that " This study addressed the effectiveness of a theory-based, multicomponent eHealth 276 intervention on fruit and vegetable intake, moderate-to-vigorous physical activity, sedentary behavior, and screen time among young children (1-3 years). Secondarily, this study investigated if there were any improvements in nutrition knowledge, attitudes, self-efficacy, and comprehensive feeding practices among their parents." However, they do not clearly discuss the reasons why their research is different from other previous studies. Authors should clearly address this issue, which allows to better understand the actual added, new information of their study, compared to knowledge already gathered by previous studies.
2. In the methods, it is necessary to further clarify how many subjects were invited to the study?
3. Additional information in methods is needed how did you defined obesity? Did you use z score or percentiles?
4. It would be interesting to see an additional Table with baseline data on age, body height, body mass, BMI and BMI SDS comparing two groups.
Author Response
Thank you very much for your contributions. The research team has implemented your feedback and recommendations. We have included our responses point-by-point to each comment. Please see the attachment of our revised manuscript.
Point 1: They do not clearly discuss the reasons why their research is different from other previous studies. Authors should clearly address this issue, which allows to better understand the actual added, new information of their study, compared to knowledge already gathered by previous studies.
Response 1: The research team provided a specific explanation in the literature context on how this study contributes to the body of knowledge on innovation. Specific page and line numbers are p. 2, lines 49-82.
Point 2: In the methods, it is necessary to further clarify how many subjects were invited to the study?
Response 2: The research team implemented the reviewer’s recommendation by revising the sentence. Specific page and line numbers are p. 3, line 101.
Point 3: Additional information in methods is needed how did you defined obesity? Did you use z score or percentiles?
Response 3: The research team implemented the reviewer's recommendation and added more detailed information to the methodology. Specific page and line numbers are p. 5, lines 156-165.
Point 4: It would be interesting to see an additional Table with baseline data on age, body height, body mass, BMI and BMI SDS comparing two groups.
Response 4: The research team provided additional information on baseline value comparisons in BMI-for-age z scores for toddlers and BMI for their parents between the study groups in the text and Table 2. The research team also added comparisons in the same parameters between the completers and dropouts in the supplementary materials file. Specific page and line numbers p. 6 & 7-8, lines 230-231, 237-238 & 242-245

Reviewer 2 Report
The Authors present original research titled: Effects of a Theory-Based, Multicomponent eHealth Intervention for Obesity Prevention in Young Children from Low-Income Families: A Pilot Randomized Controlled Study.
The manuscript shows the effectiveness of eight weeks of nutritional education on child health behaviors. The Authors analyzed parent-administered questionnaires at baseline and post-intervention. The data showed a significantly increased daily intake of fruit and vegetables and decreased screen time use in the intervention group compared to the control group. Parents improved self-efficacy and comprehensive feeding practices; however, no significant differences were observed between the study groups for changes in child outcomes (including children's physical activity and sedentary behaviors).
1. The authors proved that the nutritional education of parents influences the child's health. Obesity is an epidemy, and we know that changing lifestyle through nutritional modifications affects obesity prevalence. What is innovatory in your study?
2. What was the reason for not changing the physical activity of children? What kind of interventions might change the children's activity?
3. What clinical improvement have you observed? Did the children reduce body mass? Did some biochemical parameters been improved (glucose, lipid levels)?
4. Did you observe the correlation between Parental Educational Attainment or Household Income Level and children's positive behavioral/nutritional changes?
Author Response
Thank you very much for your contributions. The research team has implemented your feedback and recommendations. We have included our responses point-by-point to each comment. Please see the attachment, our revised manuscript.
Point 1: The authors proved that the nutritional education of parents influences the child's health. Obesity is an epidemy, and we know that changing lifestyle through nutritional modifications affects obesity prevalence. What is innovatory in your study?
Response 1: The research team implemented the reviewer's recommendation and provided a specific explanation in the literature context on how this study contributes to the body of knowledge on innovation. Specific page and line numbers are p. 2, line 49-82.
Point 2: What was the reason for not changing the physical activity of children? What kind of interventions might change the children's activity?
Response 2: The research team discussed possible reasons for the nonsignificant changes in physical activity among children. Specific page and line numbers are p. 11, line 349-358.
Point 3: What clinical improvement have you observed? Did the children reduce body mass? Did some biochemical parameters been improved (glucose, lipid levels)?
Response 3: Thank you for pointing out a lack of clinical outcomes. It has been addressed as a limitation of this study. Specific page and line numbers are p. 12, line 414-415.
Point 4: Did you observe the correlation between Parental Educational Attainment or Household Income Level and children's positive behavioral/nutritional changes?
Response 4: The research team implemented the reviewer's recommendation and provided the results of the correlational analysis as explanatory outcomes in the Supplementary Materials file. Specific page and line numbers are p. 13, line 430-431.

Round 2
Reviewer 1 Report
The authors responded to all objections.